# Solving the Quadratic Assignment Problem using Deep Reinforcement Learning

## Abstract

The Quadratic Assignment Problem (QAP) is an NP-hard problem which has proven particularly challenging to solve: unlike other combinatorial problems like the traveling salesman problem (TSP), which can be solved to optimality for instances with hundreds or even thousands of locations using advanced integer programming techniques, no methods are known to exactly solve QAP instances of size greater than 30. Solving the QAP is nevertheless important because of its many critical applications, such as electronic wiring design and facility layout selection. We propose a method to solve the original Koopmans-Beckman formulation of the QAP using deep reinforcement learning. Our approach relies on a novel double pointer network, which alternates between selecting a location in which to place the next facility and a facility to place in the previous location. We train our model using A2C on a large dataset of synthetic instances, producing solutions with no instance-specific retraining necessary. Out of sample, our solutions are on average within 7.5% of a high-quality local search baseline, and even outperform it on 1.2% of instances.

## 1 Introduction

Reinforcement learning has been used to solve problems of increasing difficulty over the past decade. Examples include AI models developed by Mnih et al. (2013) to play Atari video games and by Silver et al. (2018) to play chess and go. These successes have inspired a wave of research on using reinforcement learning to tackle hard problems.

One area of exciting progress has been in the use of reinforcement learning to solve combinatorial optimization problems. These problems are an ideal case study for reinforcement learning: they are almost always NP-hard, and are therefore among the hardest problems in computer science; at the same time, they are easy to state and experiment with. Initial work by Bello et al. (2016) tackled the traveling salesman problem (TSP), perhaps the most well-studied combinatorial optimization problem. Subsequent works by Nazari et al. (2018), Kool et al. (2018), Delarue et al. (2020), Li et al. (2021) focused on the more complex capacitated vehicle routing problem. More recently, approaches have been proposed for vehicle routing problems with time windows (Falkner & Schmidt-Thieme, 2020) and min-max routing problems (Son et al., 2023). These approaches share a *constructive* design, in which a high-quality solution is sequentially constructed; in contrast, Ma et al. (2021) and Wu et al. (2021) propose *improvement* designs, where a feasible solution is iteratively improved.

These works have made valuable progress on solving combinatorial optimization problems with reinforcement learning, relying on specially-designed attention mechanisms (Kool et al., 2018), matrix embeddings (Kwon et al., 2021) or modified training algorithms (Kwon et al., 2020). However, one drawback is that they typically do not outperform existing non-learning approaches for the specific combinatorial optimization problems they consider (Cappart et al., 2023). Such an outcome is not very surprising, since existing combinatorial solvers are the result of decades of research and problem-specific optimizations. For example, the Concorde TSP solver by Applegate et al. (2002) can easily solve TSPs with thousands of locations to optimality. Yet this outcome motivates further research into problems where reinforcement learning can bring value beyond existing algorithms.

One possible path is to look for harder problems. While most combinatorial optimization problems are NP-hard, some are more NP-hard than others. For instance, the quadratic assignment problem

(QAP) is not only NP-hard to solve exactly; it is also NP-hard to solve approximately (with a constant approximation ratio, see Sahni & Gonzalez, 1976). In the operations research literature, the QAP is often described as "one of the hardest combinatorial optimization problems" (Loiola et al., 2007) that "seems to defy all solution attempts except for very limited sizes" (Erdoğan & Tansel, 2011). This complexity means there is potential for learning approaches to make an impact.

In this paper, we present a reinforcement learning approach for the quadratic assignment problem. We first formulate it as a sequential decision problem, which we solve using policy gradient algorithms. Our approach relies on a novel double pointer network which can construct a sequence of decisions which alternates between one "side" of the assignment problem and the other.

We construct a novel combination of successful techniques in the literature on reinforcement learning for combinatorial optimization. We reformulate the quadratic assignment problem as a sequential decision problem, in order to leverage sequence-to-sequence decoding architectures (Sutskever et al., 2014; Vinyals et al., 2015), which were successfully applied by Nazari et al. (2018) for the capacitated vehicle routing problem. Additionally, we leverage an attention mechanism, first introduced by Vaswani et al. (2017) and often applied to combinatorial problems (Kool et al., 2018). We also use graph convolutional networks (Chung et al., 2014) to embed problem data. We note that our approach relies on a constructive design rather than an improvement design.

Machine learning approaches for QAP are somewhat scarce in the literature. Nowak et al. (2018) propose a supervised learning approach, training their model on previously solved problem instances. More recently, Wang et al. (2019) and Wang et al. (2020) propose novel embedding techniques to extract high-quality solutions to graph matching problems closely related to quadratic assignment. The complexity of QAP remains an obstacle, with Pashazadeh & Wu (2021) identifying particular challenges that learning approaches must overcome in order to make progress on solving the QAP.

## 2 THE QUADRATIC ASSIGNMENT PROBLEM

### 2.1 PROBLEM FORMULATION

Several versions of the QAP exist; in this work, we focus on the original formulation from Koopmans & Beckmann (1957). We are given a set of $n$ facilities, denoted by $\mathbb{F}$, and a set of $n$ candidate locations, denoted by $\mathbb{L}$; the flow from facility $i$ to facility $j$ is denoted by $F_{i,j}$, and the distance from location $k$ to location $\ell$ is denoted by $D_{k,\ell}$. If we place facility $i$ at location $k$ and facility $j$ at location $\ell$, we incur a cost of $F_{i,j} \cdot D_{k,\ell}$, representing the cost of transporting $F_{i,j}$ units of flow across the distance $D_{k,\ell}$. Let $X_{i,k}$ be the binary decision variable that takes the value 1 if we place facility $i$ in location $k$, and 0 if we do not. We can formulate the QAP as the following integer program:

$$\min \quad \sum_{i=1}^{n}\sum_{k=1}^{n}\sum_{j=1}^{n}\sum_{\ell=1}^{n} F_{i,j}D_{k,\ell}X_{i,k}X_{j,\ell} \qquad = \boldsymbol{F} \cdot \left(\boldsymbol{X}\boldsymbol{D}\boldsymbol{X}^{\top}\right) \qquad (1\text{a})$$

$$\text{s.t.} \quad \sum_{i=1}^{n} X_{i,k} = 1 \qquad\qquad \forall k \in [n] \qquad (1\text{b})$$

$$\sum_{k=1}^{n} X_{i,k} = 1 \qquad\qquad \forall i \in [n] \qquad (1\text{c})$$

$$X_{i,k} \in \{0,1\} \qquad\qquad \forall i \in [n], k \in [n]. \qquad (1\text{d})$$

Constraint (1b) ensures that each location is assigned exactly one facility, while constraint (1c) ensures that each facility is assigned exactly one location. The objective function (1a) can be written as a sum (left) or as the "dot product" of two matrices (right) — by "dot product" here we mean the sum of the elementwise product of $\boldsymbol{F}$ and $\boldsymbol{X}\boldsymbol{D}\boldsymbol{X}^{\top}$.

The QAP is difficult to solve as an integer program because of the nonlinear nature of its objective. Not only is the problem NP-hard: Sahni & Gonzalez (1976) also showed that the existence of a polynomial-time approximation algorithm with constant factor for QAP would imply that $P = NP$. In this sense, it is "harder" than many other combinatorial optimization problems. For example,

though the metric traveling salesman problem is also NP-hard, a $3/2$-approximate solution can always be obtained in polynomial time (Christofides, 1976). Indeed, it can be shown that the QAP generalizes many other combinatorial optimization problems.

## 2.2 SEQUENTIAL VIEW

In order to solve the QAP using deep reinforcement learning, we need a way to solve this combinatorial optimization problem as a sequence of decisions. We can then learn the best decision to take at each step, given the decisions taken in the past. We choose a sequential formulation in which we first select a location, then a facility to place in this location; once this pair has been selected, we choose the next location, then another facility to place in this location; and so on until each location has received a facility.

Formally, the state of the system $s_t$ at time step $t$ is represented as an alternating sequence of locations and facilities, ending at a location if $t$ is odd, and at a facility if $t$ is even. For example, at $t = 4$ we can write $s_4 = (\ell_0, f_1, \ell_2, f_3)$, where $\ell_0 \in \mathbb{L}$ and $\ell_2 \in \mathbb{L}$ are the locations selected in steps 0 and 2, while $f_1$ and $f_3$ are the facilities selected in steps 1 and 3. In this case, facility $f_1$ was placed in location $\ell_0$, facility $f_3$ in location $\ell_2$, and we are now seeking our next location.

Given this characterization of the state space, the action space straightforwardly consists of the set of remaining facilities if the last element of the sequence $s_t$ is a location, and the set of remaining locations if the last element of the sequence $s_t$ is a facility (or the sequence is empty). In our example, the action $a_4$ must be selected from the set $\mathbb{L} \backslash \{\ell_0, \ell_2\}$. We can therefore write this action as $a_4 = \ell_4$, and given this action, we deterministically transition to the state $s_5 = (\ell_0, f_1, \ell_2, f_3, \ell_4)$.

In order to complete our sequential framework, we also need to define an intermediate cost function at each step $t$:

$$
r_t(s_t, a_t) = \begin{cases} \sum_{p=0}^{\frac{t-1}{2}} \left( F_{f_{2p+1}, f_t} \cdot D_{\ell_{2p}, \ell_{t-1}} + F_{f_t, f_{2p+1}} \cdot D_{\ell_{t-1}, \ell_{2p}} \right) & \text{if } t \text{ is odd } (a_t = f_t), \\ 0 & \text{if } t \text{ is even } (a_t = \ell_t). \end{cases} \tag{2}
$$

In other words, when we place facility $f_t$ at location $\ell_{t-1}$, we incur the distance cost of transporting the flows from all *previously placed* facilities to facility $f_t$ as well as the distance cost of transporting the flows from $f_t$ to all previously placed facilities.

For simplicity, in this paper we restrict ourselves to *symmetric* QAP instances, where the flow from facility $i$ to facility $j$ equals the flow from facility $j$ to facility $i$, i.e., $F_{i,j} = F_{j,i}$ for all $i, j$. Additionally, we make the assumption that the locations are points in $\mathbb{R}^2$, and we consider the Euclidean distances between these locations. As a result, the matrix $\boldsymbol{D}$ is also symmetric, and the inner summand in (2) simplifies to $2F_{f_{2p+1}, f_t} D_{\ell_{2p}, \ell_{t-1}}$.

We emphasize that this decomposition is far from trivial. In the QAP, the cost of placing a facility in a location depends not just on the current facility and the current location, but indeed on the placements of every other facility. As a result, a sequential decomposition is less readily constructed than for other combinatorial optimization problems, For instance, the traveling salesman problem can be written as a sequence of locations ("cities") to visit, with the agent incurring the distance from the current city to the next city at each step. In comparison, our proposed sequential model is far less intuitive — a necessity given the intrinsic difficulty of the QAP.

## 3 DEEP REINFORCEMENT LEARNING MODEL

The objective of this paper is to develop an approach to learn an optimal policy $\pi^*(\cdot)$ mapping each state $s_t$ to an optimal action $a_t$. We rely on a policy gradient approach in which we directly parametrize the policy, and learn its parameters using an advantage-actor-critic (A2C) approach. We now describe the architecture of the neural network modeling our policy.

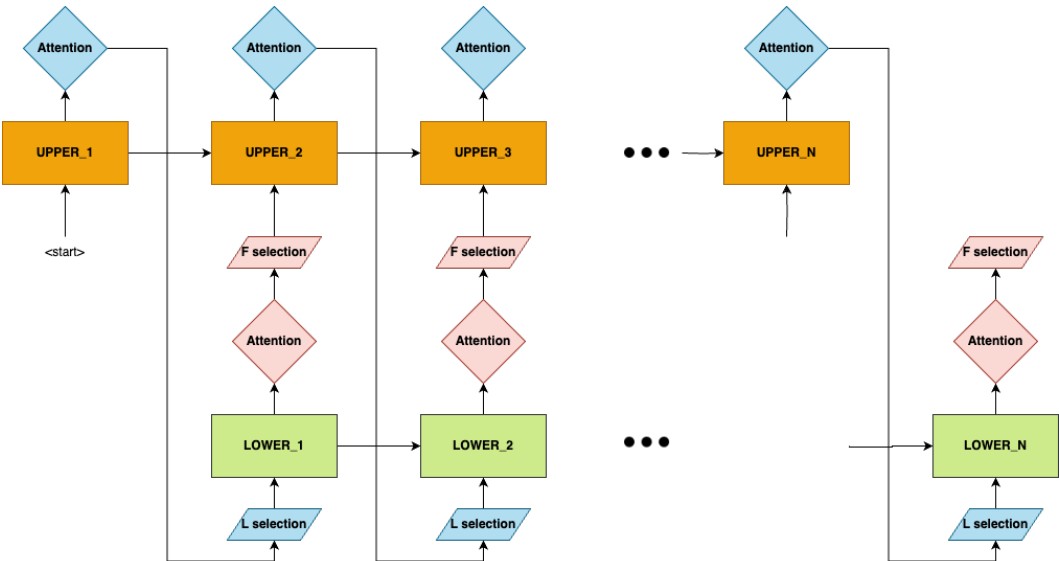

Figure 1: Diagram of double pointer network. The "upper" and "lower" GRU blocks share the same weights; since this is a decoding architecture, the output of each GRU of each type is an input to the next GRU of the same type. The "upper" pointer network selects locations, while the "lower" pointer network selects facilities.

## 3.1 EMBEDDINGS

We observe from formulation (1) that a QAP instance is uniquely specified by the $n \times n$ matrix $\boldsymbol{F}$ of flows between all facility pairs, and the $n \times n$ matrix $\boldsymbol{D}$ of distances between all location pairs. The first step of our method is to embed this data into higher-dimensional spaces.

We would like to embed each of the $n$ locations in a higher-dimensional space in a way that incorporates the notion of distance, and separately embed each of the $n$ facilities in a higher-dimensional space in a way that incorporates the notion of flow. For the locations, we start from an $2 \times n$ matrix (one column per location). We use three sequential one-dimensional convolution layers (1d-ConvNet) to embed the matrix of locations into $\mathbb{R}^{d_k \times n}$ (each location is represented as a vector of length $d_k$).

For the facilities, we take advantage of the flow matrix symmetry to represent the facilities as the nodes of an undirected complete weighted graph; the weight of edge $(i, j)$ represents the flow $F_{i,j} = F_{j,i}$. With this transformation, a natural choice of embedding is to use a graph convolutional network (GCN). By applying 3 sequential GCNs, we obtain an embedding where each facility is represented as a vector in $\mathbb{R}^{d_k}$, resulting in a $d_k \times n$ facility embedding matrix.

## 3.2 DOUBLE POINTER NETWORK

In order to produce an alternating sequence of locations and facilities, we develop a novel double pointer network architecture. Like Nazari et al. (2018), we observe that the QAP inputs (locations and facilities) are not ordered to begin with: as a result, unlike traditional sequence-to-sequence architectures, an encoding step is not necessary. We therefore construct a decoder which takes in an arbitrary start token and consists of $2n$ alternating blocks of type $U$ ("upper") and $L$ ("lower"). We need two different chains of pointer network units because, unlike routing problems which can be formulated as sequentially selecting elements ("cities") from a single set, we must pair elements from one set with elements from another set. Our double pointer network aims to generate such an alternating sequence.

Pointer blocks of type $U$ take as input either the start token, or the embedding of the last selected facility, and output a vector of length $n$ containing the probabilities of next selecting each location $\ell_k$. Pointer blocks of type $L$ take the embedding of the last selected location as input, and output

a vector of length $n$ containing the probabilities of placing facility $i$ in this location. The output of each pointer block of type $U$ is also an input to the next block of type $U$, and the output of each pointer block of type $L$ is also an input to the next block of type $L$. A diagram of the decoding pointer network is shown Figure 1.

In order to output these action probabilities, each pointer block consists of a Gated Recurrent Unit (GRU, see Chung et al., 2014, for implementation details). The output of the GRU, of dimension $d_k$, is then passed through an attention layer, which we describe in the next section.

### 3.3 ATTENTION

In order to convert the output of GRU of type $U$ or $L$, we introduce an attention layer. Informally, this layer performs the "pointing" part of the pointer network, by comparing the output of the GRU (a vector in $\mathbb{R}^{d_k}$) to the vectors embedding each location (or facility), and producing a vector of $n$ probabilities specifying the next location (or facility) to select.

Attention can refer to several related but different mechanisms. Our approach is closest to the one used by Nazari et al. (2018): we compute an attention and context vector as intermediate steps before producing a vector of output probabilities. However, we remove the nonlinear $\tanh$ activation function; we also use the output of the pointer block instead of the hidden state (noting that the two are the same for a single-layer RNN). Without loss of generality, we describe the attention procedure for the upper pointer network (location selection).

The attention layer consists of three major steps. The first step is to compute an $n$-dimensional attention vector. We first consider the matrix of all embedded locations, denoted by $\boldsymbol{L} \in \mathbb{R}^{d_k \times n}$, whee each column is a $d_k$-dimensional location embedding. We obtain an *extended* location matrix by appending the $d_k$-dimensional GRU output to the end of each column of $\boldsymbol{L}$, and denote this extended location matrix by $\tilde{\boldsymbol{L}} \in \mathbb{R}^{2d_k \times n}$. Let $\boldsymbol{v}_a \in \mathbb{R}^{d_i}$ and $\boldsymbol{W}_a \in \mathbb{R}^{d_i \times 2d_k}$ be a vector and matrix of trainable parameters. The attention vector is then given by

$$\boldsymbol{a}^\top = \text{softmax}\left(\boldsymbol{v}_a^\top f(\boldsymbol{W}_a \tilde{\boldsymbol{L}})\right),$$

where $f(\cdot)$ designates an arbitrary activation function. Nazari et al. (2018) use $f(\cdot) = \tanh(\cdot)$; in our implementation we simply use the identity function $f(\boldsymbol{x}) = \boldsymbol{x}$.

The second step is to compute a context vector $\boldsymbol{c} \in \mathbb{R}^{d_k}$, which is obtained as a weighted combination of the location embeddings, where the weights are specified by the attention vector, i.e. $\boldsymbol{c} = \boldsymbol{L}\boldsymbol{a}$. Finally, the third step closely resembles the first step, using the context vector as input and producing the output probability vector $\boldsymbol{o}$ as

$$\boldsymbol{o}^\top = \text{softmax}\left(\boldsymbol{v}_o^\top f(\boldsymbol{W}_o \hat{\boldsymbol{L}})\right),$$

where $\boldsymbol{v}_a \in \mathbb{R}^{d_i}$ and $\boldsymbol{W}_a \in \mathbb{R}^{d_i \times 2d_k}$ are trainable parameters, and $\hat{m L} \in \mathbb{R}^{2d_k \times n}$ is obtained by appending the context vector $\boldsymbol{c}$ to each column of the embedding matrix $\boldsymbol{L}$.

The output vector $\boldsymbol{o}$ is $n$-dimensional and sums to one. It represents the probability of selecting each of the $n$ locations as our next action; in other words, at step $t$, the probability of taking action $a_t = \ell_k$ is specified by $o_k^t$, where $\boldsymbol{o}^t$ designates the output vector of pointer block $t$. For lower pointer blocks, we replace the matrix $\boldsymbol{L}$ of location embeddings with the matrix of flow embeddings in all three steps. We train two sets of attention parameters: one shared between all the upper pointer units, and one shared between all the lower pointer units.

### 3.4 TRAINING

We train the model our model via Advantage Actor-Critic (A2C, see Mnih et al., 2016). A2C is a policy gradient algorithm which requires a critic network to estimate the problem value function. Given a value function estimate $V(s_t)$ for any state $s_t$, the advantage $A_t$ of taking a particular action $a_t$ in state $s_t$ is defined as

$$A_t(a_t) = r_t(s_t, a_t) + \gamma V(s_{t+1}) - V(s_t).$$

Given a sample path $(s_0, a_0, s_1, a_1, \ldots, s_T)$, we can use the policy gradient theorem to compute that the gradient of our loss function $\nabla_\theta J(\theta)$ with respect to trainable parameters $\theta$ is proportional

to $\sum_{t=0}^{T-1} \nabla_\theta \log \pi_\theta(s_t) A_t(a_t)$. We therefore define our training objective as

$$l(\theta) = \sum_{t=0}^{T-1} \nabla_\theta \log \pi_\theta(s_t) A_t(a_t) + \alpha \sum_{t=0}^{T-1} (A_t(a_t))^2 + \beta \sum_{t=0}^{T-1} H(\boldsymbol{o}_t),$$

where $\alpha$ is a parameter controlling the importance of the critic training loss in the overall training loss, and $\beta$ is a regularization parameter; for $\beta > 0$, the regularization term seeks to maximize the entropy of our current policy to encourage exploration.

The critic network has a very simple architecture: a multi-layer perceptron (MLP) with an input dimension of $2n^2 + 2n$ (the flattened flow and distance matrices, concatenated with the sequence of indices of previously selected locations and facilities). Our MLP includes two hidden layers with 512 and 1024 neurons, respectively.

## 4 RESULTS

### 4.1 DATA AND SETUP

We generate a dataset of QAP instances from the following distribution: locations are sampled uniformly at random in the two-dimensional unit square. The flow from facility $i$ to facility $j$ is sampled uniformly at random from $[0, 1]$. We add the obtained random matrix to its transpose to yield a symmetric flow matrix, then set the diagonal elements to zero. For most experiments, we use a training set of up to 100,000 instances and evaluate results on a test set of 1,000 different instances from the same distribution.

We train for 20 epochs, using a batch size of 50 instances. For regularization, we add a dropout probability of 0.1. For our largest training data set size (100,000 instances), our typical training time per epoch on 2 NVIDIA A100 GPUs is 20 minutes and 1 hour for $n = 10$ and $n = 20$, respectively.

To reduce both noise in the results and overfitting, we cache the trained model at every epoch and evaluate it on a separate validation dataset of 1000 QAP instances. We report all metrics using the best cached model, not the last obtained model.

When evaluating sample paths during training, we sample from the action selection distribution to ensure exploration. At test time, we deterministically select the action with the highest probability. We also implement a beam search procedure where we continuously maintain the top 5 or 10 highest-probability sample paths so far; once the terminal state is reached, we keep the path with the lowest cost on the particular instance under study.

### 4.2 BASELINE

The most rigorous optimization approach for combinatorial problems like QAP is integer programming, using a specialized solver like Gurobi (Gurobi Optimization, LLC, 2023). However, as mentioned previously, integer programming approaches can be very slow for the QAP, so a more tractable baseline is desirable. We choose a simple swap-based local search heuristic (see Algorithm 1 in the appendix). Given an initial feasible solution, the heuristic greedily swaps the locations of facility pairs as long as a cost-reducing swap exists; when no such swap exists or when we reach an iteration limit of 1000, the heuristic terminates.

Our key success metric is the *percentage gap* between the objective value $c_{\text{sol}}$ of our solution and the objective value of the solution $c_{\text{swap}}$ obtained via the swap heuristic, i.e., $(c_{\text{sol}} - c_{\text{swap}})/c_{\text{swap}}$.

### 4.3 MODEL PERFORMANCE

We first evaluate the performance gap of our model as compared to the swap heuristic for QAP instances of size $n = 10$ and $n = 20$. We compare the "greedy" version of our RL model (where the highest-probability action is selected deterministically at each step) with beam-search versions with beam sizes 5 and 10. Results are shown in Table 1. On average, our reinforcement learning model produces results within 7.5% of the swap combinatorial heuristic. Performance is quite consistent, with a worst-case (95th percentile) gap below 15% in most cases. We note that performance remains

consistent across varying instance sizes: for $n = 20$, the average performance gap stays roughly the same as for $n = 10$, but the 95th percentile gap decreases markedly. Similarly, the fraction of test instances with a gap below 10% increases significantly from $n = 10$ to $n = 20$; however, for $n = 10$ the RL method is the best one (negative gap) on up to 1.5% of instances, while we never outperform the baseline for $n = 20$.

Table 1: Performance gap of RL approach for varying QAP instance size.

| $n$ | Method | Performance gap (%) | | Fraction (%) of instances | |
| --- | --- | --- | --- | --- | --- |
| | | Average | 95th percentile | Gap $\leq 10\%$ | Gap $\leq 0\%$ |
| | RL-Greedy | 9.09 | 16.39 | 59.9 | 0.5 |
| 10 | RL-Beam 5 | 7.64 | 14.55 | 74.1 | 1.0 |
| | RL-Beam 10 | 7.14 | 13.94 | 78.2 | 1.4 |
| | RL-Greedy | 8.19 | 11.75 | 82.0 | 0.0 |
| 20 | RL-Beam 5 | 7.67 | 11.15 | 87.7 | 0.0 |
| | RL-Beam 10 | 7.46 | 10.85 | 90.1 | 0.0 |

We also perform runtime comparisons for our models and the combinatorial baselines, with results shown in Table 2. All runtime experiments (evaluating our RL model, or calling Gurobi or the swap solver) are performed on a single laptop (Macbook Pro with M1 chip). We first observe that Gurobi requires several orders of magnitude more time than any other method, timing out after two minutes on the majority of instances for both $n = 10$ and $n = 20$. For $n = 10$, it does produce substantially better solutions than any other method, but for $n = 20$ it is significantly outperformed by the swap heuristic.

We also find that even though our RL model does not quite match the performance of the swap solver in terms of objective, it achieves its results with a significant reduction in running time (up to an order of magnitude without using beam search). These runtime results are a reminder of the value of a reinforcement learning algorithm that can solve previously unseen combinatorial problems by simply evaluating a neural network (admittedly, a complex one). Additionally, we observe that for $n = 20$, the RL approaches are a thousand times faster than Gurobi, yet almost match its performance (within 2%). Finally, even though the swap baseline produces the best solutions, we observe that its runtime increases by a factor of 10 from $n = 10$ to $n = 20$, while the runtime of our RL approaches increases by a factor of 2. This linear trend is valuable since it means RL has the potential to tackle even larger problem instances.

Table 2: Runtime and performance of RL approach versus standard combinatorial methods. Results are averaged over 10 instances for Gurobi and 1000 instances for other methods, and are presented with standard errors. Note the standard errors for RL-Greedy are somewhat overestimated due to batching. For tractability, we set a timeout of 120s for Gurobi, and observe that it times out on the majority of instances

| Method | $n = 10$ | | $n = 20$ | |
| --- | --- | --- | --- | --- |
| | Average cost | Runtime (s) | Average cost | Runtime (s) |
| Gurobi | $20.41 \pm 0.70$ | $119.4 \pm 0.6$ | $95.90 \pm 2.28$ | $120.3 \pm 0.009$ |
| Swap | $21.29 \pm 0.10$ | $0.02 \pm 0.0001$ | $91.02 \pm 0.25$ | $0.2 \pm 0.001$ |
| RL-Greedy | $23.21 \pm 0.10$ | $0.002 \pm 0.003$ | $98.48 \pm 0.27$ | $0.004 \pm 0.005$ |
| RL-Beam 5 | $22.91 \pm 0.10$ | $0.03 \pm 0.0001$ | $97.98 \pm 0.27$ | $0.06 \pm 0.0001$ |
| RL-Beam 10 | $22.80 \pm 0.10$ | $0.05 \pm 0.0005$ | $97.78 \pm 0.27$ | $0.1 \pm 0.0004$ |

## 4.4 ABLATION STUDIES AND ADDITIONAL RESULTS

We now study the impact of various components if our model via ablation studies. We first consider the value of the attention layer, which is often a critical component of RL frameworks for

Table 3: The value of attention. Ablation study conducted on problem size $n = 10$. The train and test sets both have 1000 instances, and we train for 100 epochs.

| Decoder architecture | Method | Performance gap (%) Train | Test |
|---|---|---|---|
| MLP | RL-Greedy | 10.86 | 10.82 |
| | RL-Beam 5 | 9.64 | 9.61 |
| | RL-Beam 10 | 9.18 | 9.18 |
| Attention | RL-Greedy | 9.62 | 9.58 |
| | RL-Beam 5 | 8.33 | 8.26 |
| | RL-Beam 10 | 7.79 | 7.77 |

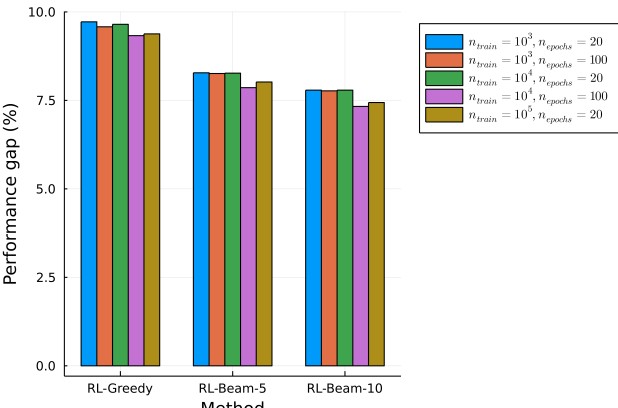

Figure 2: Effect of training dataset size and number of epochs on out-of-sample performance. Gap measured relative to the swap baseline.

combinatorial optimization. To conduct this ablation study, we replace every attention block with a simple one-layer MLP, transforming the $d_k$-dimensional GRU output into an $n$-dimensional probability vector. We compare performance in Table 3. We observe that the attention layer is responsible for approximately 1.5 percentage points of performance improvement, or about a 20% improvement in relative terms. Note that in this study, we report the performance of the final model (and not the best cached model according to the validation set).

This result is not very surprising: attention mechanisms are an essential component of many modern deep learning architectures for their ability to relate components of the problem (e.g., words in a sentence) to each other. This architecture allows us to learn a policy that can be tailored to each instance and therefore can generalize well out of sample.

Finally, we compare the out-of-sample results of our model as we vary the training dataset size and the number of training epochs. The results for $n = 10$ are shown in Figure 2. We observe that increasing the number of epochs seems to have a larger impact on the greedy model performance than increasing the number of training samples. The beam search models benefit about equally from more epochs as they do from more training data points.

## 5 DISCUSSION

The results presented in Section 4 are encouraging. They demonstrate that reinforcement learning approaches for QAP have potential, obtaining performance gaps of under 7.5% on problem instances of size $n = 10$ and $n = 20$. We also empirically verify the complexity of solving the QAP exactly, with our Gurobi solver timing out after two minutes, usually with a suboptimal solution (especially for $n = 20$).

In contrast with Pashazadeh & Wu (2021), we therefore find that there is tremendous potential to solve the QAP with reinforcement learning. At the same time, challenges remain: first and foremost, to further increase the performance of RL approaches to more closely match both Gurobi and the swap baseline. More generally, there is a fundamental gap between classic optimization approaches, which provide not only a solution but also a certificate of its optimality, and RL approaches, which only provide a solution. If the goal is only to provide solutions, then we feel that a particularly intractable problem like QAP is a good choice for further study — indeed the results in Table 2 show that RL evaluation is even faster than the swap heuristic — the persistence of this effect at higher problem sizes would make a strong case for further research in this direction.

If the goal of RL approaches for combinatorial optimization is to truly compete with established optimization methods, then it is of interest to design methods that produce not just high-quality solutions but also lower bounds that prove their quality — an exciting direction for future research.

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
