# A APPENDIX

## A.1 IMPLEMENTATION DETAILS

We provide additional implementation details for model-building and training. Our training, validation, and testing sets are randomly generated with the seeds 42, 43, and 44, respectively. This also ensures that our testing set is the same across all experiments in the ablation studies. The main results in Table 1 are obtained by training on 2 AI100 GPUs, while the ablation study results are all obtained by training on 4 RTX6000 GPUs. Inference is always performed on a Macbook Pro CPU. Model parameters are initialized using the Xavier method (Glorot & Bengio, 2010).

We also include a full pseudocode for the swap-based baseline in Algorithm 1.

---

**Algorithm 1** Greedy Swap Heuristic for QAP

---

**Require:** flow matrix ($F$) and distance matrix ($D$)
1: $X \leftarrow I_n$ {start from the identity matrix}
2: $Cost(X) = \sum_{i=1}^{n} \sum_{j=1}^{n} F_{ij}(XDX^\top)_{ij}$
3: $iters \leftarrow 0$
4: **while** $iters \leq 1000$ **do**
5:    $X_{prev} \leftarrow X$
6:    **for** all possible row swap neighbors of $X$ **do**
7:       $X' \leftarrow$ row swap neighbor of $X$
8:       **if** $Cost(X') < Cost(X)$ **then**
9:          $X \leftarrow X'$
10:       **end if**
11:    **end for**
12:    $iters = iters + 1$
13:    **if** $X_{prev} = X$ **then**
14:       **break**
15:    **end if**
16: **end while**
17: **return** $X, Cost(X)$

---

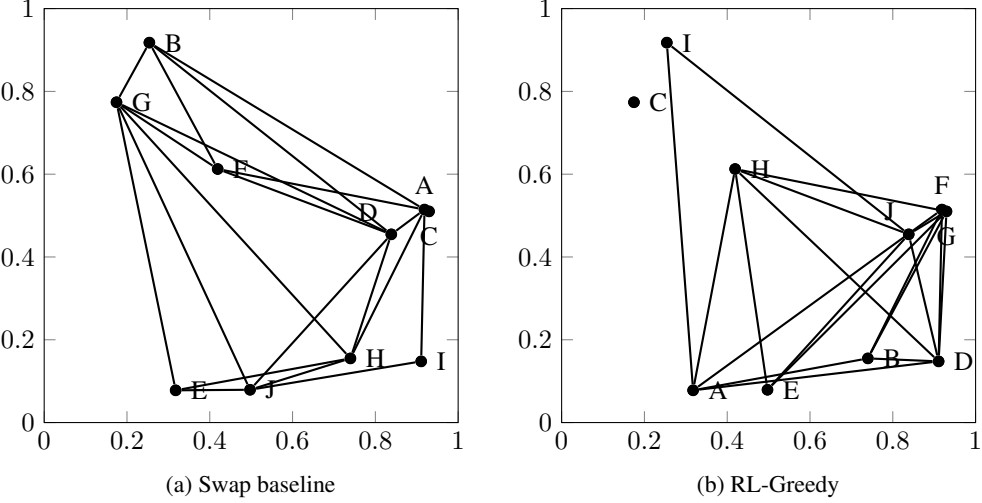

(a) Swap baseline            (b) RL-Greedy

Figure 3: Example of a QAP instance where the RL greedy method outperforms the swap baseline. Facilities are labeled from $A$ to $J$ and placed in their assigned locations. Edges between facilities represent the 20 largest flows between any pair of facilities.

## A.2 CASE STUDY: RL-GREEDY VS SWAP

We now present a more detailed visualization of a test instance where we outperform the swap heuristic ($n = 10$). On this instance, we obtain a total cost of 20.4, versus 21.3 for the swap baseline. In Figure 3, we contrast the swap and RL solutions on this particular instance, by placing the ten facilities (labeled from $A$ to $J$) in the locations assigned by each method. Edges between facilities represent the 20 largest flows between any pair of facilities; as a result, the total length of these edges is an approximation of the cost of our solution.

Interestingly, we observe that none of the 20 largest flows in this instance involve facility $C$, and only two involve facility $I$. The RL method accordingly places these two facilities in the top left corner, relatively far from the other facilities. In contrast, the swap algorithm places both $C$ and $I$ very close to the main cluster of facilities, which is visibly inefficient. We hope that a better understanding of these patterns can help us further improve the performance of our RL methods.