# OpenReview forum: "Solving the Quadratic Assignment Problem With Deep Reinforcement Learning"
_ICLR.cc/2024/Conference — Submitted to ICLR 2024_

### Official Review · Reviewer_Gzbd · 2023-10-22

**Soundness:** 2 fair
**Presentation:** 2 fair
**Contribution:** 1 poor
**Rating:** 3
**Confidence:** 4

**Summary:**

In this paper, the authors investigated the Quadratic Assignment Problem (QAP) solving using deep reinforcement learning with double pointer networks, where an upper pointer network selects locations and a lower pointer network selects facilities. And the proposed method is evaluated on synthetic QAP instances.

**Strengths:**

1- This paper is well motivated as the QAP has been rarely studied using DRL;
2- It seems the proposed method works on the self-generated dataset.

**Weaknesses:**

1- It seems that from the DRL based heuristic perspective, the objective function in QAP does not make difference compared with the ones in ILP, both of which are just captured by a reward in DRL. In this sense, the QAP in this paper is almost the same as the vehicle and customer assignment in VRPs with linear objectives, such as
[a] Deep Reinforcement Learning for Solving the Heterogeneous Capacitated Vehicle Routing Problem. IEEE T Cybernetics;
[b] Learning to Solve Vehicle Routing Problems with Time Windows through Joint Attention. Arxiv;
[c] Solving NP-hard Min-max Routing Problems as Sequential Generation with Equity Context. Arxiv.
Those works also involve sequentially or parallelly selecting vehicles and customers, while using (advanced) Transformers rather than pointer network.

2- Besides, the Transformers in 'Learning Improvement Heuristics for Solving Routing Problems. TNNLS' outputs a probability matrix, which could also be tailored to the probability of facility and location pair with proper masking.

3- The evaluation is quite simple, which solely focuses on one single problem with randomly generated instances. And the baselines are also insufficient and not strong enough.

Overall, the method falls short of novelty, and the evaluation is inadequate, which is below the standard of an ICLR publication.

**Questions:**

Please see the above weakness.

---

> ### Author Response · Authors · 2023-11-22
>
> Thank you for the feedback. While we focus on a facility location motivation, it is true that the quadratic assignment problem can be viewed as a generalization of several other combinatorial optimization problems such as the traveling salesman problem and some vehicle routing problems – indeed, we feel this is motivation for developing a DRL methodology to solve it. We have added the suggested references to our literature review, and we are working to implement these suggestions (as well as those made by other referees) to improve performance and narrow the gap with our baseline methods.

---

### Official Review · Reviewer_DVdt · 2023-10-29

**Soundness:** 3 good
**Presentation:** 3 good
**Contribution:** 3 good
**Rating:** 5
**Confidence:** 5

**Summary:**

This paper delves into the end-to-end application of Deep RL for solving the QAP. Because the QAP necessitates assignments between a set of facilities and locations, the authors introduce a novel autoregressive model that sequentially selects a location and then a facility (to make a pair), and then repeats this process until the assignment is done. The results are promising. The derived solutions are, on average, within 7.5% of those generated by a heuristic method.

**Strengths:**

This paper represents the pioneering effort to tackle the QAP using a neural network-based approach, marking an important milestone in the field. The paper is well-structured, with contents presented in a manner that's easy to follow and understand.

**Weaknesses:**

The results are too weak for ICLR publication. While one could argue that the considerable optimality gap might be attributed to the intrinsic complexity of the QAP, it's evident that a notable portion arises from the authors' dependence on a neural net model and the training methods that closely mirror the early contributions of Bello et al. that might not be the best fit.

To draw a parallel, when the pointer network was initially applied to solve the TSP in an end-to-end manner, there existed an optimality gap of around 7% for 100-node TSPs. However, with subsequent refinements in models and training techniques, that gap has been narrowed to almost 0% now.

It's plausible that similar methodological advancements could significantly benefit the QAP approach presented here. I'd recommend reconsidering the use of the Critic network for such combinatorial optimization tasks, especially given the challenges of predicting the final outcome midway through solution construction.

A few references that could be helpful are:
[model]
Matrix Encoding Networks for Neural Combinatorial Optimization (Kwon, et al.)
Learning to Iteratively Solve Routing Problems with Dual-Aspect Collaborative Transformer (Ma, et al.)
[RL method]
POMO: Policy Optimization with Multiple Optima for Reinforcement Learning (Kwon, et al.)

**Questions:**

None

---

> ### Author Response · Authors · 2023-11-22
>
> Thank you for your feedback. We really appreciate the constructive suggestions as to how we can improve model performance and narrow the gap with existing methods. We have added references to the suggested works, and are working to integrate some of these ideas into our model. In particular, we are thinking about how to improve the performance of our critic network, as well as how to select a more problem-specific attention mechanism.

---

### Official Review · Reviewer_tFs4 · 2023-10-29

**Soundness:** 1 poor
**Presentation:** 2 fair
**Contribution:** 1 poor
**Rating:** 1
**Confidence:** 4

**Summary:**

This paper proposes an RL method to solve quadratic assignment problems. The authors reformulate the original QAP as a seq2seq problem. However, the paper is not easy to follow. The motivations and contributions of the work are not clear.  The details of the method are missing. The experimental evaluation is not good enough. Therefore, I think the paper is under the bar of ICLR in its current form.

**Strengths:**

The topic of using RL to solve the quadratic assignment problem is interesting.

The authors reformulate the original QAP as a seq2seq problem.

**Weaknesses:**

The motivation and contribution of the work is not clear.

The paper is not easy to follow.

The literature review is not good enough. A lot of classic literature is missing.

The experimental evaluation is not enough. Only Gurobi and swap-based local search heuristic is compared. Which makes the paper less convincing.

The size of QAP used in the experiment is quite small.

The performance is quite poor, which cannot outperform simple heuristics such as SWAP.

**Questions:**

Please see the weeknesses.

---

> ### Author Response · Authors · 2023-11-22
>
> Thank you for your feedback. We have expanded the literature review to include some additional relevant references suggested by the referees, as well as clarified the motivation and contribution in the introduction. Regarding the evaluation, we feel that adding further baselines is difficult, since to our knowledge there are no other RL frameworks for QAP. We are working on improving performance relative to the swap heuristic by making tweaks to our model architecture, following the suggestions of other reviewers. We are also working to improve scalability by reducing model size where it is possible to do so without compromising performance.

---

### Official Review · Reviewer_jXD8 · 2023-11-01

**Soundness:** 3 good
**Presentation:** 4 excellent
**Contribution:** 3 good
**Rating:** 6
**Confidence:** 4

**Summary:**

The Quadratic Assignment Problem (QAP) is an NP-hard problem with significant challenges, especially for larger instances. The paper proposes using deep reinforcement learning (DRL) to address the QAP, introducing a novel double pointer network to tackle the Koopmans-Beckman formulation of QAP. The method is trained using A2C on synthetic datasets, and its performance is benchmarked against a swap-based local search heuristic.

**Strengths:**

Using deep reinforcement learning to address the QAP is an innovative method, setting it apart from traditional optimization techniques. The double pointer network alternates between selecting locations and facilities, providing a dynamic solution approach. The model is trained on a large dataset of synthetic instances, making it robust and generalizable.

**Weaknesses:**

The method is tested primarily for QAP instances up to size 20, highlighting a potential scalability concern. While the DRL approach shows promise, there is still a performance gap when compared to the swap heuristic and the Gurobi solver. Unlike traditional optimization methods which provide a solution and its optimality certificate, this DRL approach only gives a solution.

**Questions:**

Why was the swap-based local search heuristic chosen as the baseline?
Is the performance gap mentioned equivalent to the standard duality or MIP gap commonly used in optimization? If not, how do they differ? The obscurity in the definition makes it difficult to assess the performance of the new method.
How does the DRL method scale with larger problem sizes, especially when compared with traditional algorithms?
Why is training limited to 20 minutes and 1 hour for n = 10 and n = 20? How will performance improve if the training time is increased?

---

> ### Author Response · Authors · 2023-11-22
>
> Thank you for the feedback. Regarding baselines: we chose the swap heuristic because it provides near-optimal solutions (within about 1% of the provable optimum found by Gurobi) in a very short time. The gap provided is therefore a (nearly tight) lower bound and good approximation of the standard MIP gap. Regarding scalability, we are working on reducing the number of model parameters to generalize to larger sizes; while scalability is indeed a concern, this is even more true for Gurobi, which is not able to solve the problem in hours for sizes larger than 20. Regarding training time, we clarify that the reported training times of 20 minutes and 1 hour were per training epoch.

---

### Meta-Review · Area_Chair_LoPC · 2023-12-08

**Metareview:**

The authors develop an RL based approach for solving a specific class of combinatorial optimization problems, the quadratic assignment problem, and perform evaluations comparing to a local search heuristic. Reviewers agree that the evaluation is not sufficient to justify the value of the method, and hence I recommend rejection.

**Justification For Why Not Higher Score:**

RL approach for combinatorial optimization that ignores much prior literature and only evaluates on a single problem.

**Justification For Why Not Lower Score:**

N/A

---

### Decision · Program_Chairs · 2024-01-16

Reject